# *TP53* p.R337H Germline Variant among Women at Risk of Hereditary Breast Cancer in a Public Health System of Midwest Brazil

**DOI:** 10.3390/genes15070928

**Published:** 2024-07-16

**Authors:** Tatiana Strava Corrêa, Paula Fontes Asprino, Eduarda Sabá Cordeiro de Oliveira, Ana Carolina Rathsam Leite, Luiza Weis, Maria Isabel Achatz, Claudiner Pereira de Oliveira, Renata Lazari Sandoval, Romualdo Barroso-Sousa

**Affiliations:** 1Hospital Sírio-Libanês, Centro de Oncologia de Brasília, Brasília 71635-610, DF, Brazil; tatiana.scorrea@hsl.org.br (T.S.C.); anarathsam@gmail.com (A.C.R.L.); rsandoval.med@gmail.com (R.L.S.); 2Instituto de Ensino e Pesquisa do Hospital Sírio Libanês, São Paulo 01308-060, SP, Brazil; pasprino@mochsl.org.br (P.F.A.); luiza_weis@yahoo.com.br (L.W.); miachatz@mochsl.org.br (M.I.A.); 3Faculdade de Medicina, Universidade de Brasilia (UNB), Brasília 70910-900, DF, Brazil; eduardasabaa@gmail.com; 4Hospital Materno Infantil de Brasília (HMIB), Asa Sul 70203-900, DF, Brazil; 5Instituto Hospital de Base do Distrito Federal (IHB-DF), Brasília 70330-150, DF, Brazil; 6DASA Oncology, Hospital Brasília, Brasília 71681-603, DF, Brazil; 7Unidade de Genética do Hospital de Apoio de Brasília, Secretaria de Saúde, Brasília 70684-831, DF, Brazil; claudinero@yahoo.com.br

**Keywords:** breast cancer, hereditary cancer, Li–Fraumeni syndrome, *TP53* p.R337H

## Abstract

Despite the high prevalence of *TP53* pathogenic variants (PV) carriers in the South and Southeast regions of Brazil, germline genetic testing for hereditary breast cancer (HBC) is not available in the Brazilian public health system, and the prevalence of Li-Fraumeni syndrome (LFS) is not well established in other regions of Brazil. We assessed the occurrence of *TP53* p.R337H carriers among women treated for breast cancer (BC) between January 2021 and January 2022 at public hospitals of Brasilia, DF, Brazil. A total of 180 patients who met at least one of the NCCN criteria for HBC underwent germline testing; 44.4% performed out-of-pocket germline multigene panel testing, and 55.6% were tested for the p.R337H variant by allelic discrimination PCR. The median age at BC diagnosis was 43.5 years, 93% had invasive ductal carcinoma, 50% had estrogen receptor-positive/HER2 negative tumors, and 41% and 11% were diagnosed respectively at stage III and IV. Two patients (1.11%) harbored the p.R337H variant, and cascade family testing identified 20 additional carriers. The *TP53* p.R337H detection rate was lower than that reported in other studies from south/southeast Brazil. Nonetheless, identifying *TP53* PV carriers through genetic testing in the Brazilian public health system could guide cancer treatment and prevention.

## 1. Introduction

Li-Fraumeni syndrome (LFS) is a cancer predisposition syndrome caused by germline pathogenic variants (PVs) in the tumor suppressor gene *TP53*. Approximately 90% of the carriers develop tumors by the time they reach 60 years of age [1]. Adrenocortical carcinoma, sarcoma, brain cancer, and breast cancer are among the most common cancers in the phenotypic LFS tumor spectrum [2,3]. Breast cancer (BC) is the most prevalent tumor type in women with LFS [4].

LFS is rare worldwide, affecting 1 in 5000–20,000 individuals [2,5], but its impact in Brazil is relevant because of a founder effect, which is responsible for the high prevalence of *TP53* c.1010G>A (p.Arg337His or p.R337H) carriers [6]. Studies of cohorts from the Southern region found a prevalence of 1 in 300 live births harboring the *TP53* p.R337H variant [7,8].

The prevalence of the *TP53* p.R337H variant in BC cohorts in Brazil has been previously estimated to range from 0 to 12%, depending on the geographic region [8,9,10,11,12,13]. Differences in the geographical distribution of this variant have increased interest in the determination of the frequency of p.R337H in the population of the Federal District of Brazil, a central area of the country, where families from different regions converge.

A recent study of patients with BC evaluated in a private hospital in the Federal District of Brazil showed that 2.7% were p.R337H carriers (6/224); however, while approximately 80% of Brazilian have access to health care only in the public service, the study cohort mostly comprised patients with health insurance [14]. Considering the lack of germline testing for patients using the public health system in the Federal District, the frequency of p. R337H in an underserved population remains of interest.

This exploratory study aimed to describe the frequency of *TP53* p.R337H as well as the clinical and pathological characteristics of patients diagnosed with BC treated in the Brazilian Public Health System (SUS) in the Federal District of Brazil who were stratified as high risk for hereditary breast cancer (HBC).

## 2. Materials and Methods

### 2.1. Study Design

This observational cross-sectional study was conducted at a public institution in the Federal District. We recruited patients with a history of invasive BC of any stage and any clinical subtype, who were undergoing treatment or in follow-up and presented at least one of the National Comprehensive Cancer Network (NCCN) criteria for HBC, considering version 1.2020 [15] (Appendix A). These patients were offered to participate in the study and consented to undergo testing for the *TP53* p.R337H variant specified in the study protocol. Patients who chose to perform a multigene panel testing at commercial laboratories at their own expense were also included in the study. Additionally, all these patients underwent genetic counseling. First-degree family members with positive *TP53* p.R337H index cases were offered genetic counseling and family variant testing. Patients with missing clinicopathological BC data or those who did not undergo germline testing were excluded.

Blood samples were collected for those patients who were going to test only the *TP53* p.R337H variant.

The primary outcome was to describe the detection rate of the *TP53* p.R337H in these patients. We also aimed to describe the percentage of patients who met the revised Chompret criteria [2,16] (Appendix A) and identify family members carrying the *TP53* p.R337H variant based on the index case.

This study was approved by the Research Ethics Committee of Sírio-Libanês Hospital and the Brazilian Research Ethics Committee (CAAE: 31299320.2.0000.5461). All the participants provided informed consent.

### 2.2. Genotyping for TP53 p.R337H

DNA was extracted from peripheral blood samples using a Wizard^®^ Genomic DNA Purification Kit (Promega, Madison, WI, USA). Genotyping for *TP53* p.R337H was performed using a Custom TaqMan^®^ SNP Genotyping Assay, performing allelic discrimination (Applied Biosystems, Thermo Fisher Scientific, Pleasanton, CA, USA).This assay is specific to detecting the *TP53* p.R337H variant, and the presence of other variants in *TP53* or in other genes could not be evaluated for those patients who performed only this genetic test.

Allelic discrimination analysis was performed in a 7300 (Applied Biosystems) real-time PCR equipment using a Custom TaqMan™ SNP Genotyping Assay (Thermo Fisher). Fwd Primer: CCTCCTCTGTTGCTGCAGATC, Rev Primer: CCTCATTCAGCTCTCGGAAC, Probe 1-GGTGAGCGCTTCGAG (WT_VIC/MGB-NFQ), Probe 2-CGTGAGCACTTCGAG (Mut_R337H_ FAM/MGB-NFQ). Reactions were prepared for a total volume of 12 uL, using 1× TaqMan genotyping master mix, 1× TaqMan Primers + Probes mix and 50 ng of DNA. After the Pre-read step, amplification was performed by 60 °C for 1 min, 95 °C for 10 min and 50 cycles of 95 °C for 15 s/60 °C for 90 s, followed by 60 °C for 1 min. Post-read analysis was defined using control samples.

### 2.3. Sociodemographic and Clinical Variables

During the genetic counseling consultation, sociodemographic data were collected: self-declared race and birthplace. Clinical and pathological data were retrospectively reviewed using the electronic medical records of all study participants.

### 2.4. Statistical Analysis

Values are expressed as medians and percentiles for non-normally distributed continuous variables and as means and standard deviations for normally distributed continuous variables. Categorical data are presented as absolute values and percentages and were tested using Fisher’s exact test, where applicable. Analyses were performed using JASP version 0.14.1.0 software.

### 2.5. Sample Calculation

An estimated 730 new cases of BC are estimated for 2020 in the Federal District (INCA). Of these cases, 70% will be treated in the public service; therefore, the estimated N is 511 patients to be screened per year diagnosed with breast cancer in public hospitals in the DF. With the majority being followed up at IHB-DF, we expect at least 300 patients (60%) with breast cancer per year at this hospital. For the primary objective, the detection rate of PV p.R337H, for a population of 300 women with breast cancer treated in public hospitals in the Federal District from January 2021 to January 2022, considering an average prevalence of 4.8% for PV R337H of TP53 obtained from previous studies [10], for a sampling error of 2% and a confidence interval of 95%, it was estimated that 178 patients would be needed to compose the sample.

## 3. Results

### 3.1. Clinical and Pathological Characteristics of the Population

A total of 245 women with BC were screened for this study. Of these, 180 patients met the study criteria. The median age at BC diagnosis was 43.5 years (range: 18–78 years). Among the participants, 75 (42%) were brown. The median interval between BC diagnosis and genetic counseling was one year, ranging from 0.1 to 12 years. The demographic, clinical, and pathological characteristics of the patients are summarized in Table 1.

Regarding invasive BC features, most were invasive ductal carcinomas (93%), whereas the remainder were invasive lobular carcinomas (6%) and other types (1%). Most tumors (50%, n = 90) were estrogen receptor-positive (ER+)/HER2-negative, 45 (25%) were HER2-positive, and 45 (25%) were triple-negative. A total of 21 women (9.5%) had bilateral breast tumors, including both synchronous and metachronous cases. In terms of initial breast cancer staging, 94 patients (52%) were at stage III or IV at the time of BC diagnosis. At the time of genetic counseling, 31 patients (17%) had metastatic disease.

Most patients were native to the Midwest (47%) or Northeastern (35%) regions of Brazil. Furthermore, 5% of patients hailed from the North, 1% from the South, and 11% from the Southeast (Figure 1).

### 3.2. Revised Chompret Criteria for Li-Fraumeni Syndrome

Forty patients (40/180; 22%) met the revised Chompret criteria for LFS. Among these 40 patients, 40% (n = 16) met the criteria for BC diagnosis before 31 years of age, and 60% (n = 24) met the criteria for BC diagnosis at or before 45 years of age, along with a positive family history of sarcoma, brain cancer, or adrenocortical carcinoma. Among those patients who were found to harbor the p.R337H variant, 5% (2/40) met the revised Chompret criteria for LFS.

### 3.3. TP53 p.R337H Detection Rate

Among the 180 included patients, 100 underwent allelic discrimination testing and 80 underwent germline multigene panel testing. We identified two patients (1.1%) with the p.R337H variant, one through variant testing and the other through a multigene panel. Both patients met the NCCN HBC criteria, as they were diagnosed with BC at or before 45 years of age.

One patient (Figure 2, R337H.01) met the revised Chompret criteria for LFS because of a BC diagnosis at 28 years of age; that is, she met the criteria of juvenile breast cancer (<31 years). She was submitted to two molecular tests (a multigene out-of-pocket test and the specific one for p.R337H offered by the study). This patient was born in the Federal District, with a family from the Midwest region, presented with stage IIIA invasive ductal carcinoma, histological grade 2, ER+/HER2-negative, and Ki67 > 14%. The patient underwent neoadjuvant chemotherapy, followed by bilateral mastectomy and adjuvant hormone therapy. Familial cascade testing revealed variant segregation through the maternal side of the family.

Another patient (Figure 3, R337H.02) was diagnosed with BC at 43 years of age and had a family history of cancer. This history included a nephew diagnosed with brain cancer at four years of age, as well as seven close relatives diagnosed with BC: four sisters, one niece, one paternal cousin, and one paternal aunt.

The patient was born in the Federal District, but her family was from the Southeast region. She was diagnosed with de novo stage IV, ER+/HER2-negative invasive ductal carcinoma (grade 2).

Genetic counseling and family variant testing were extended to 31 relatives, revealing that 19 individuals were carriers of the p.R337H variant.

### 3.4. Patients Who Underwent Germline Genetic Panel Testing

Among 80 patients who underwent out-of-pocket multigene panel testing, 17 (21.2%) carried pathogenic variants (PV) in cancer susceptibility genes, including 12 (15%) patients with *BRCA1/BRCA2* PV, two (2.5%) with *TP53* PV, one with p.R337H and one with non-R337H *TP53* PV (p.W146X), along with 3 (3.7%) in genes not associated with HBC (Appendix A in the Appendix A).

Regarding the timing of genetic testing, 37 (46%) patients underwent genetic testing before the surgical planning, whereas 43 (54%) underwent testing only during follow-up or at the time of disease recurrence.

## 4. Discussion

Our results suggest a lower prevalence of *TP53* p.R337H among women with BC in the Brazilian Midwest public health system compared to other Brazilian studies. Nonetheless, a significant number of family members harboring this variant were identified through cascade genetic testing, reinforcing the potential impact of genetic testing as a preventive approach.

We found 1.1% *TP53* p.R337H carriers among the studied population, characterized by public health users, with a median age at BC diagnosis of 43.5 years. The population was enriched with individuals from the Midwest and Northeast regions of the country. This finding is in concordance with studies from the Northeast that reported lower frequencies of *TP53* p.R337H carriers, ranging from 0% to 0.6% [12,13]. In contrast, Giacomazzi et al. found 12% *TP53* p.R337H carriers among Southern women diagnosed with BC at or before 45 years of age [9].

Although haplotype studies and Brazilian colonization history have indicated a common Portuguese ancestor as an explanation for the higher concentration of *TP53* p.R337H carriers in the Southeastern region of the country [17,18], the largest Brazilian BC genotyping study, which included 1663 BC patients unselected by age at BC onset or family history of cancer, mostly from the South/Southeast of Brazil (70%), had a detection rate of p.R337H carriers of 1.6% [19]. These data suggest that regional colonization differences and population selection criteria affect the *TP53* p.R337H detection rate, as one can see in Table 2.

It is also important to point out other clinical characteristics of the cohort. Similar to the AMAZONA study [20], in which 80.8% of the patients received treatment through the public health system, we observed that a substantial proportion of the patients were diagnosed with locally advanced or metastatic breast cancer (52% in stages III or IV). Several Brazilian studies have indicated an association between the socioeconomic status of women and their ability to undergo recommended preventive exams, such as Pap smears and mammography [21,22]. Therefore, a lack of resources in a given population is associated with delayed cancer diagnosis at more advanced stages.

**Table 2 genes-15-00928-t002:** Studies on the prevalence of the p.R337H variant in context.

Reference	N	Inclusion Criteria	Brazil’s Region	*TP53* R337H, n (%)
[8]	750	Healthy women	South	2 (0.3)
[9]	815	BC, not selected	South and Southeast	70 (8.6)
[23]	120	BC, with high-risk HBC (NCCN)	Southeast	3 (2.5)
[24]	95	High-risk HBC (NCCN)	Southeast	5 (5.3)
[25]	805	BC, not selected	South	19 (2.36)
[26]	224	BC, with high-risk HBC (NCCN)	Midwest	6 (2.7)
[19]	1663	BC, with high-risk HBC (NCCN)	South and Southeast (70%)	26 (1.6)
[12]	173	BC, with high-risk HBC (NCCN)	Northeast	1 (0.6)
[13]	355	BC, with high-risk HBC (NCCN)	Northeast	0
This study	180	BC, with high-risk HBC (NCCN)	Midwest	2 (1.1)

The primary clinical utility of germline genetic testing for hereditary cancer lies in identifying individuals at a higher risk of developing cancer, thereby providing an opportunity for effective cancer screening and implementing risk reduction strategies. However, owing to the limited resources and constraints of the Brazilian public health system, patients often face restrictions accessing most screening examinations and may be required to cover the costs themselves. The Toronto protocol, a follow-up protocol for LFS carriers, has been shown to improve early cancer detection and survival rates [5,27]. According to Frankenthal et al., the Toronto protocol is cost-effective for LFS carriers in Brazil [28]. Annual whole-body MRI, a component of the protocol, detects neoplasms in 9% of asymptomatic patients. Unfortunately, a lack of specific public information renders the complete protocol unavailable to users of the Brazilian public health system.

This study had some limitations. First, only patients alive at the time of enrollment were included in this study, which may represent a survivorship bias. Second, the cohort consisted mostly of patients from two public hospitals in the Federal District, which serve 80% of the population who need oncologic care and do not have health insurance in the region; this fact can represent a socioeconomic bias. Other economic barriers may have influenced patient recruitment, since there was no financial support for enrollment or transportation expenses. In addition, one hundred patients were only tested for the specific *TP53* p.R337H; among these, we may have missed carriers of other *TP53* non-p.R337H carriers and carriers of PV in other breast cancer susceptibility genes.

## 5. Conclusions

The detection rate of the *TP53* p.R337H variant was lower than that reported in other Brazilian studies in Southeastern Brazil. Despite this, our findings draw attention to the LFS carriers overlooked by the Brazilian public healthcare system. The implementation of hereditary cancer screening through genetic testing and preventive interventions is urgently required for this underserved population.

## Figures and Tables

**Figure 1 genes-15-00928-f001:**
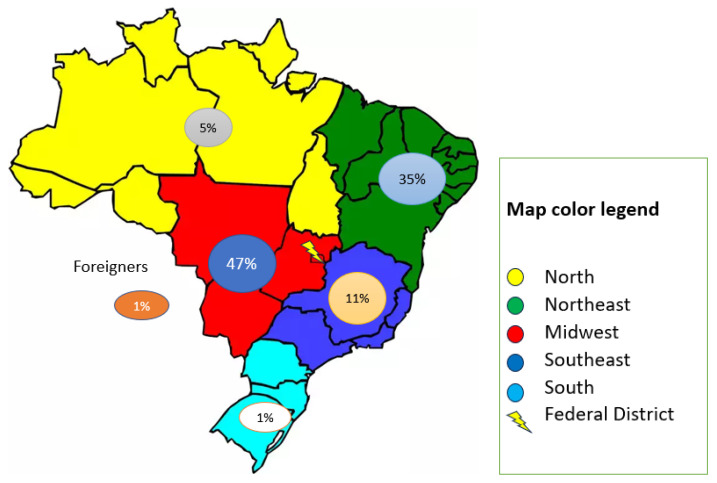
Map of Brazil showing the central position of Brasilia (Federal District) and the region of birth of the study participants.

**Figure 2 genes-15-00928-f002:**
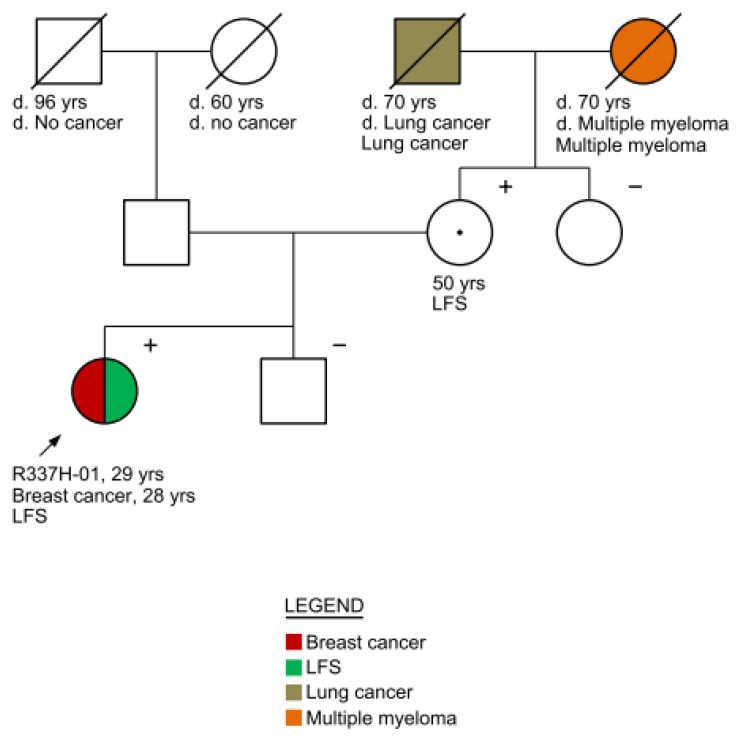
Pedigree of R337H.01.

**Figure 3 genes-15-00928-f003:**
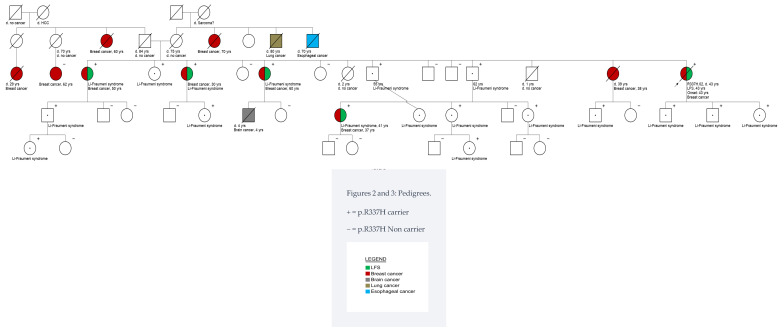
Pedigree of R337H.02.

**Table 1 genes-15-00928-t001:** Clinical and pathological characteristics of the cohort.

Characteristics	N = 180	
Age	43.5 years	(18–78)
Median follow-up	1 year	(0.1–12)
Self-defined ethnicity	Brown	76 (42%)
	White	68 (38%)
	Black	36 (20%)
	Asian/Indigenous	0
Histology	Invasive ductal carcinoma	166 (93%)
	Invasive lobular carcinoma	11 (6%)
	Other *	3 (1%)
Immunohistochemistry	Estrogen receptor-positive, HER2-negative	90 (50%)
	HER2-positive	45 (25%)
	Triple-negative	45 (25%)
Ki67	≥14%	155 (86%)
	<14%	25 (14%)
Staging	I	16 (9%)
	II	70 (39%)
	III	74 (41%)
	IV	20 (11%)

* Other: mucinous carcinoma, papillary carcinoma.

## Data Availability

The original contributions presented in the study are included in the article/Appendix A, further inquiries can be directed to the corresponding author.

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
