# Peer review of "TP53 p.R337H Germline Variant among Women at Risk of Hereditary Breast Cancer in a Public Health System of Midwest Brazil"

_genes, 2024, doi:10.3390/genes15070928_

Round 1

Reviewer 1 Report

Comments and Suggestions for Authors

Major Comments

Since the study focuses on a topic that has been well researched and established, it is necessary to state what is novel and significant in the abstract and reiterate it in the conclusion. It is quite clear that germline testing for the R337H in the Brazilian population is important, given the founder effect in the population. However, it is important to state the key take-aways clearly.

  1. Please clarify throughout the text where you specify "TP53 carriers" that you mean carriers of the TP53 R337H variant.
  2. Line 99 - Reiterate the criteria met here or clarify in the Methods section about the "study criteria", since methods talks about NCCN and Chompret criteria and it is not clear what the study criteria comprise.
  3. Line 159 - does the "two TP53 PV" mean they had the R337H variant? It is important to clarify since you follow up with non-R337H variants.
  4. Line 130 - met the "HBC" Criteria - This terminology was not used before in the text and it is difficult to tell if it is referring to the NCCN criteria. It is important to stay consistent with terminology.
  5. It is not clear what statistical significance tests were performed or needed for this study, so may be the Methods can be modified to what is specific for this study.

Minor Comments:

  1. Abstract uses HBC and BC as abbreviations. This needs to be clarified to emphasize the distinction
  2. Line 73 is incomplete
  3. Line 76: Use Bougeard et al, 2015 reference for revised Chompret criteria
  4. Lines 133-135 needs to be rephrased for clarity
  5. Presence and absence of variant should be indicated in the legend for Fig 2
  6. Figure 2 Legend LFS is specified twice, can be a bigger font size
  7. Reorder references
Comments on the Quality of English Language

The manuscript may need some English language editing with incomplete sentences, confusing phrasing and missing words.

Reviewer 2 Report

Comments and Suggestions for Authors

Tatiana Strava Correa et al. investigate the prevalence of TP53 p.R337H carriers among women diagnosed with breast cancer in Brasília, Brazil, over a one-year period. The study enrolled 180 patients who met at least one of the National Comprehensive Cancer Network (NCCN) criteria for hereditary breast cancer (HBC). Genetic testing was conducted using either a germline multigene panel or a p.R337H-specific allelic discrimination PCR. The study found a lower prevalence of the p.R337H variant compared to earlier studies in the southern and southeastern regions of Brazil. The findings suggest that incorporating genetic testing into the Brazilian public health system could be beneficial for guiding cancer treatment and prevention.

This research fills a crucial gap in the understanding of genetic testing and hereditary cancer variants within the Brazilian public health system. By focusing on a less-studied region, the study broadens the knowledge base and underscores the importance of expanding genetic testing services to improve cancer care outcomes. I suggest that this manuscript is suitable for publication in Cells.

Author Response

Thank you very much for taking the time to review this manuscript.

Reviewer 3 Report

Comments and Suggestions for Authors

This manuscripts reports the analysis of the TP53 p.R377H pathogenic variant in a breast cancer  cohort from the midwest Brazil. Indeed, this variant has been reported as associated to a founder effect in other papers analyzing women from the southern of Brazil.

Among the 180 enrolled patients just 2 carried the mutations, highlighting discrepant findings respect to previously reported data. However cascade family test allowed to identify several carriers to be admitted to primary prevention protocols.

About half of the study subject underwent private multigenerational panel testing, thus other mutation were identified; BRCA genes being the most mutated as expected.

I think that this aspect should be better developed and discussed in the manuscript. Even if the starting hypothesis is to verify the frequency of the TP53 p.R377H, we cannot consider that BRCA genes should be evaluated as first line test in the presence of criteria allowing the suspicion for a hereditary BC. Indeed, in the patients that performed the single-mutation assay, we cannot exclude the presence of other pathogenic variants in the same the TP53 or in other genes. All these aspects need to be mentioned, even if (or taking into account) the public health system in Brazil don't provide genetic analyses.

Specific point:

- section 2.2. Genotyping should be better described indicating for example probes sequences and the instrument used.

- line 121: should be "patients, 40% (n=40)".

- Figures 2 and 3 should have a more comprehensive legend explaining color code. Moreover patients ID is reported only in the legend and must be reported also in the main text.

Discussion and Conclusion should be modified taking into account the above-mentioned issues.

In the supplementary table 1 is better to report the rs rather than the Clinvar ID.

Reviewer 4 Report

Comments and Suggestions for Authors

The article titled " TP53 p.R337H germline variant among women at risk of hereditary breast cancer in a public health system of Midwest Brazil" by Tatiana Strava Correa et al. provides valuable insights. However, there are several areas that require attention:

1. The authors should provide their own justification for the study, as previous publications have already explored the relevance of the topic. Examples of such publications include articles in PubMed, Hered Cancer Clin Pract. 2014 Mar 13;12(1):8. doi: 10.1186/1897-4287-12-8. PLoS One. 2014 Jun 17;9(6):e99893. doi: 10.1371/journal.pone.0099893. Biomolecules. 2022 Apr 27;12(5):640. doi: 10.3390/biom12050640. Cancer Lett. 2007 Jan 8;245(1-2):96-102. doi: 10.1016/j.canlet.2005.12.039; among others. Consequently, the study does not provide any innovative information.

2. The study only includes public hospitals of Brasilia, DF, Brazil with the occurrence of TP53 p.R337H carriers among women treated for breast cancer, so its generalizability to other populations or entire country is uncertain.

3. The sample sizes is relatively small with only 180, which may limit the statistical power of the analysis and the ability to draw conclusions about these the prevalence of TP53 p.R337H carriers in this population

4. Authors mentioned that 44.4% of patients underwent out-of-pocket germline multigene panel testing, while 55.6% were tested for the p.R337H variant by allelic discrimination PCR. This difference in testing methods could show bias and affect the comparability of the outcomes

5.  Authors did not provide data on the characteristics of the patients who underwent testing but did not meet the NCCN criteria for HBC

6. The authors should include limitations and future perspectives of the study. Authors should discuss the potential challenges or implications of implementing such testing on a larger scale including the financial and logistical constraints of the public health system

Comments on the Quality of English Language

Moderate editing of English language required

Round 2

Reviewer 1 Report

Comments and Suggestions for Authors

No comments to add. All my suggestions were considered and incorporated. 

Reviewer 3 Report

Comments and Suggestions for Authors

The Authors have answered to almost all my previous comments.

Reviewer 4 Report

Comments and Suggestions for Authors

Accept in present form

Comments on the Quality of English Language

 Minor editing of English language required